# A Hypertonic Seawater Nasal Irrigation Solution Containing Algal and Herbal Natural Ingredients Reduces Viral Load and SARS-CoV-2 Detection Time in the Nasal Cavity

**DOI:** 10.3390/jpm13071093

**Published:** 2023-07-03

**Authors:** Ioannis Pantazopoulos, Athanasios Chalkias, Angeliki Miziou, Michalis Spanos, Efrosyni Gerovasileiou, Erasmia Rouka, Konstantinos Gourgoulianis

**Affiliations:** 1Department of Emergency Medicine, Faculty of Medicine, University of Thessaly, 41500 Larissa, Greece; michaelspan1988@gmail.com; 2Department of Respiratory Medicine, Faculty of Medicine, University of Thessaly, 41500 Larissa, Greece; kellymiz95@yahoo.com (A.M.); gerovasileiou@yahoo.com (E.G.); kgourg@uth.gr (K.G.); 3Department of Anaesthesiology, Faculty of Medicine, University of Thessaly, 41500 Larissa, Greece; thanoschalkias@yahoo.gr; 4Institute for Translational Medicine and Therapeutics, University of Pennsylvania Perelman School of Medicine, Philadelphia, PA 19104, USA; 5Outcomes Research Consortium, Cleveland, OH 44195, USA; 6Faculty of Nursing, University of Thessaly, 45550 Larissa, Greece; errouka@uth.gr

**Keywords:** nasal irrigations, hypertonic saline, hypertonic seawater, SARS-CoV-2, COVID-19, viral load

## Abstract

Nasal irrigation is thought to decrease the viral load present in the nasal cavity. Our aim was to assess the effect of a hypertonic seawater solution [with algal and herbal natural ingredients (Sinomarin^®^)] on the viral load of nasopharynx in patients hospitalized with severe COVID-19 pneumonia. We conducted a prospective, randomized, controlled trial from June 2022 to December 2022. We allocated 56 patients with COVID-19 pneumonia into two groups (28 in each group)—the hypertonic seawater group [nasal irrigations with a hypertonic seawater solution (Sinomarin^®^) every 4 h for 16 h per day, for two consecutive days] and the control group (no nasal irrigations). A second nasopharyngeal swab was collected 48 h after the baseline nasopharyngeal swab (8 h after the last wash in the hypertonic seawater group) to estimate the SARS-CoV-2 viral load as determined by cycle threshold (Ct) values. In the hypertonic seawater group, the mean Ct values significantly increased two days after the initial measurement [ΔCt 48−0 h = 3.86 ± 3.03 cycles, *p* < 0.001 (95%CI: 2.69 to 5.04)]. No significant differences in the Ct values were observed in the control group [ΔCt 48−0 h = −0.14 ± 4.29, *p* = 0.866 (95%CI: −1.80 to −1.52)]. At follow-up, 17 patients from the hypertonic seawater group had negative test results compared to only 9 patients from the control group (*p* = 0.03). Nasal irrigations with a hypertonic seawater solution containing algal and herbal natural ingredients significantly decreased nasopharyngeal viral load and the detection time of SARS-CoV-2 in the nasal cavity.

## 1. Introduction

Since the beginning of the coronavirus disease 2019 (COVID-19) pandemic, several variants of severe acute respiratory syndrome coronavirus 2 (SARS-CoV-2) have emerged, accelerating the spread of the virus [1] and posing a serious threat to public health [2]. It is widely accepted that the major entry point of the viruses is the nasal passage [3]. The nasal mucosa serves as a reservoir for viral replication, and the load of SARS-CoV-2 virus in the upper respiratory tract is thought to be a proxy for risk of transmission [4,5]. Therefore, any strategy or method which reduces the SARS-CoV-2 viral load in the nasal mucosa could decrease transmission and severe disease.

Nasal saline irrigations have been proposed as an effective method for reducing nasal viral load and preventing local replication in the upper respiratory tract [6]. While the benefit of nasal saline irrigations has been well established, optimal saline tonicity is still debated. A recent meta-analysis evaluating nasal rinses with both isotonic and hypertonic saline solutions concluded that hypertonic saline irrigations are more beneficial for sino-nasal diseases [7]. Another study demonstrated that sulphated polysaccharides bind tightly to the S-protein of SARS-CoV-2 interfering with its binding on the host tissues [8]. Notably, fucoidans, which are sulphated polysaccharides derived from marine algae, are capable of disabling SARS-CoV-2, thus preventing its entry and infection [9].

Consequently, we hypothesized that nasal irrigations with hypertonic saline comprising sulphated polysaccharides could be a cost-effective and easily accessible approach to decrease nasal viral load and reduce viral transmission. To test this hypothesis, we designed a randomized controlled trial to investigate the effect of nasal irrigations with a hypertonic seawater solution containing sulphated polysaccharides and other herbal ingredients on the viral load in the nasopharynx of patients hospitalized with COVID-19 pneumonia.

## 2. Materials and Methods

### 2.1. Study Design

The current study constitutes a prospective, randomized, controlled, pragmatic study performed at a reference University Hospital in Larissa, Greece. The study was conducted during a seven-month period, from June 2022 to December 2022, and was performed according to the requirements of the Declaration of Helsinki [10]. The study was approved by the Ethics Committee of the local hospital (13149/20-5-2022), and the protocol was registered in ClinicalTrials.gov (NCT05729204) and can be found in full at https://clinicaltrials.gov/study/NCT05729204 (accessed on 24 June 2023).

### 2.2. Eligibility Criteria

Individuals were eligible if they met pre-specified inclusion criteria as follows: [1] adult patient (≥18 years old) hospitalized specifically for COVID-19; [2] confirmed SARS-CoV-2 infection as determined by RT-PCR test of nasopharyngeal samples; and [3] severe COVID-19 pneumonia in need of hospital admission as per the 4-category NIH clinical severity scale [11]. We excluded patients with a positive SARS-CoV-2 test admitted for non-COVID-19-related reasons, patients using intranasal sprays for at least two weeks prior to enrolment, patients who had undergone sino-nasal surgery within three months before enrolment, patients with sinusitis, patients unable to perform nasopharyngeal washes and individuals participating in other trials.

### 2.3. Randomization

Patients received the standard of care for COVID-19 [12] and were randomized into two groups. The method for randomization used was the sequentially numbered, opaque, sealed envelopes. In the hypertonic seawater group, patients received nasal irrigations with a hypertonic (2.3% NaCl) seawater solution containing two different algae types [brown algae (Undaria pinnatifida), blue-green algae (Spirulina platensis)] as well as essential oils of Mentha spicata and Eucalyptus globulus and Thymus vulgaris extract (Sinomarin^®^ Plus Algae Cold & Flu Relief, Gerolymatos International SA, Krioneri, Greece). The Sinomarin^®^ medical device utilizes continuous flow diffusion, allowing continuous solution flow as long as the nozzle is pressed, thus providing efficient cleansing. Patients allocated into the control group received no further treatment apart from standard care. All patients were hospitalized in negative pressure isolation rooms as per the COVID-19 treatment protocol. Written individual informed consent at the time of randomization was obtained from each participant or next-of-kin in case the participant was unable to provide consent. The study follows the CONSORT reporting guidelines for randomized clinical trials [13].

### 2.4. Interventions

A nasopharyngeal swab for SARS-CoV-2 nucleic acid detection was obtained from all patients at admission (baseline—zero hour) and was deposited into a sterile bottle containing virus transport medium (10 mL tube with 3 mL medium, Biobase, Biodustry, Jinan, Shandong, China). Patients who were randomized into the hypertonic group were trained on the technique of nasal irrigation as follows: blow nose before irrigation; gently tilt their neck forward; slight tilt of head to one side; insert the nozzle in the nostril parallel to the nasal septum; press firmly to squirt the solution; return to the upright position to allow the solution to work for some time; repeat in the other nostril. In case any solution ended up in their mouth, patients were advised to spit it out. The nozzle was washed with warm water and was wiped dry after each use. A 100 mL Sinomarin^®^ bottle was then provided to each patient to perform nasal irrigation in both nostrils every 4 h for 16 h per day for two consecutive days.

At 48 hours after the baseline nasopharyngeal swab and 8 h after the last nasal wash, another nasopharyngeal swab was obtained for viral load measurement. The time point of 48 h was chosen to evaluate the immediate effect of nasal irrigation, while providing sufficient time for the mucociliary clearance of the hypertonic seawater (8 h) before the second nasopharyngeal swab was taken. This was necessary because nasal irrigation could physically remove the virus. All nasopharyngeal swabs were obtained from the same nostril for each patient, and they were performed by the treating physician who was blinded to allocation.

### 2.5. Measurement of Nasal Viral Load

The Mag-DEA^®^ Dx SV commercial kit, operated by the magLEAD^®^ 12gC instrument (Precision System Science Co., Matsudo City, Chiba, Japan), was used to extract viral RNA from 400 μL of each nasopharyngeal sample. To detect SARS-CoV-2, a commercial kit (Direct SARS-CoV-2 Real-Time PCR kit, Vircell, Granada, Spain) was used that targeted the E and N genes, which had a threshold limit of detection of 3.5 copies per reaction for both genes. An endogenous internal control (RNase *p* gene region) was used for the analysis of biological samples (Direct SARS-CoV-2 Real-Time PCR kit, Vircell, Granada, Spain). A sample was considered positive for SARS-CoV-2 when the cycle threshold (Ct) values for both the E and N genes were less than 40, according to the manufacturer’s recommendations. The Ct valhues represent the number of PCR cycles required for the SARS-CoV-2 viral RNA to become detectable, and higher N gene Ct values indicated lower amounts of SARS-CoV-2 viral RNA.

### 2.6. Study Objectives

The primary objective was to investigate the effect of nasal irrigations with Sinomarin^®^ on the viral load of the nasopharynx in patients hospitalized with COVID-19 pneumonia. The secondary objective was to examine if this effect influences escalation to high flow nasal oxygen or non-invasive ventilation, time of detection of SARS-CoV-2 in the nasal mucosa, admission to intensive care unit (ICU) and mortality.

### 2.7. Data Collection

We collected baseline demographic data such as age, sex, nationality, body mass index, smoking status and date of admission. We also collected data on relevant comorbidities, Charlson comorbidity index, disease-related symptoms, status of vaccination, day of illness based on symptom onset, status of oxygenation (PaO_2_/FiO_2_) and medications administered. Furthermore, outcomes and potential adverse effects related to the use of hypertonic seawater solution were recorded. All patients were followed to hospital discharge, ICU admission or death. Those that were discharged were re-evaluated 14 days later, and nasal viral load was measured using RT-PCR.

### 2.8. Statistical Analysis

Statistical analyses were performed using the Statistical Package for Social Sciences (IBM Corp. Released 2017. IBM SPSS Statistics for Macintosh, Version 25.0. Armonk, NY, USA: IBM Corp.). Descriptive statistics were employed to present the baseline characteristics of the patients. Means and standard deviations (SD) were reported for data with normal distributions, while the medians and interquartile ranges (IQR) were used for data with non-normal distributions, as per the Shapiro–Wilk test. Paired sample t-tests and independent sample t-tests were implemented, as appropriate, to evaluate the mean Ct cycles between groups and between 0 and 48 h. The Fisher’s exact test was implemented to identify differences in the percentages of categorical data. 

A power analysis was performed using the G*Power program (Version: 3.1) for sample size estimation, based on data from our previous study [6]. Using an alpha error = 0.05, a power of 0.90 and an allocation ratio of 1, the required sample size was 26. Initially, 40 individuals were planned to be included in the study to account for a 50% drop-out rate, with the aim of increasing statistical power.

## 3. Results

During the study period, 93 consecutive Caucasian patients with COVID-19 pneumonia were screened for eligibility, and 56 were finally enrolled in the study (28 in each group) (Figure 1). Mean age was 63.6 ± 13.8 years, and 33 (58.9%) were male. Vaccination status and duration of illness at randomization did not differ between groups as illustrated in Table 1. However, more patients reported altered sense of smell or taste at enrolment in the hypertonic seawater group, although the difference was not statistically significant.

### 3.1. Changes in Viral Load via Nasopharyngeal Swab Assessment

Mean baseline N target Ct values for intranasal SARS-CoV-2 were 21.2 ± 5 and 21.2 ± 4.47 cycles [*p* = 0.52 (95% CI: −2.5–2.58)] in the hypertonic seawater and control group, respectively.

In the hypertonic seawater group, mean N gene Ct values significantly increased 48 h after the baseline measurement [ΔCt 48−0 h = 3.86 ± 3.03 cycles, *p* < 0.001 (95% CI: 2.69–5.04)], suggesting a decline in SARS-CoV-2 viral load by 18.2% (Figure 2). Adherence was 100% in the hypertonic seawater group.

On the other hand, no significant change was observed in mean N gene Ct values in the control group [ΔCt 48−0 h = −0.14 ± 4.29, *p* = 0.866 (95% CI: −1.80 to −1.52); Figure 2]. The between groups difference in mean Ct values 48 h after admission and nasal irrigations was 4 cycles (*p* < 0.001, 95% CI: −6.82 to −1.10). 

Nasal irrigations resulted in a significant decrease in nasopharyngeal viral load indifferent of the vaccination status. On the other hand, a significant increase in nasopharyngeal viral load was observed in the unvaccinated patients of the control group. Changes in Ct values according to vaccination status are shown in Table 2.

### 3.2. Follow Up

No statistically significant difference was identified in the need for respiratory support escalation with high flow nasal cannula or non-invasive ventilation (0 vs. 2) and admission to the ICU (0 vs. 1) in the hypertonic seawater and control group, respectively. At 14 days post-hospital discharge, 17 patients in the hypertonic seawater group had negative RT-PCR tests vs. 9 patients in the control group (*p* = 0.03).

### 3.3. Adverse Events

No negative effects were observed in the majority of patients in the hypertonic seawater group except for 3 individuals who reported mild nasal irritation. On the other hand, one individual in the control group required admission to the ICU and finally died, whereas none of the patients in the hypertonic seawater group were admitted to the ICU or died.

## 4. Discussion

In this prospective, randomized, controlled, pragmatic trial, nasal irrigation with hypertonic seawater solution containing natural ingredients from algae and herbs significantly decreased viral load within 48 h compared to standard care treatment. The decrease in viral load was not affected by the vaccination status. At 14 days post-discharge, more patients in the treatment group tested negative for COVID-19 compared to controls. Taking into consideration the biology and replication of SARS-CoV-2, the present study clearly suggests that hypertonic seawater is a safe and effective treatment for reducing viral load in the upper respiratory tract of hospitalized patients with COVID-19 pneumonia.

Infection from SARS-CoV-2 primarily occurs in the nasal mucosa. The latter is characterized by abundant blood vessels, mucinous glands and serous glands that create a humid environment, as well as by increased expression of angiotensin converting enzyme-2 (ACE2) receptors [14,15,16]. The coronavirus typically disrupts the ciliated epithelium causing ciliary dyskinesia that impairs mucociliary clearance and damages the respiratory epithelium, even in asymptomatic individuals [16]. Therefore, prioritizing the protection of the upper respiratory tract and mucosa is imperative for public health.

Nasal washes physically remove mucus along with the particulate material embedded in it by disrupting the viscous surface layer. The addition of saline in nasal solutions increases hydration in the deeper aqueous layer, boosting the underlying ciliary beat frequency and decreasing local inflammatory mediators [7]. This is especially useful during viral respiratory infections, which cause mucociliary dysfunction and mucostasis due to the inflammatory response [7]. Moreover, higher salt concentrations can cause an immediate and dose-dependent hindrance in the configuration of the ACE-2 receptor, impairing the binding affinity of SARS-CoV-2 [17]. Of note, laboratory studies have shown that sodium chloride (NaCl) disrupts the charged state of the plasma membrane and affects the amount of adenosine diphosphate (ADP) and triphosphate (ATP) [18], reducing the available energy of the infected cells [18]. This ultimately prevents virus replication. In addition, it has been demonstrated that NaCl has the ability to block 3Clpro, which is a viral enzyme responsible for controlling multiple phases of replication [19], as well as the host protease furin which is utilized by the SARS-CoV-2 to trigger the rapid fusion of its spikes [20]. Previous research has revealed that NaCl has the capability to obstruct the proteolytic activity of furin [21]. Saline also modifies the activity of myeloperoxidase in nasal and pharyngeal epithelial or phagocytic cells, which enhances the production of antiviral molecules, such as hypochlorous acid [22].

Early in the course of the pandemic, various authors suggested the use of nasal irrigation as a potential public health measure against the novel virus [23], based mainly on studies in patients with other upper respiratory tract infections. A study which was frequently cited to support the aforementioned claim was a pilot randomized controlled trial by Ramalingam et al. [24]. In their work, the authors studied the effects of hypertonic saline nasal irrigation and gargling (HSNIG) versus standard care on previously healthy adults within 48 h of onset of an upper respiratory tract infection (URTI). The trial aimed to evaluate recruitment, acceptability, symptom duration and viral shedding of multiple viruses, including coronavirus. The authors showed that individuals practicing HSNIG had a shorter duration of illness and used fewer over-the-counter medications, resulting in reduced viral shedding and a lower rate of household viral transmission. Although the difference in viral load between the HSNIG and the control groups was not significant, the trial was not designed to specifically detect such differences. With the emergence of the pandemic, the same authors also performed a post hoc analysis of their data focusing only on patients who had been infected with coronaviruses [25]. They concluded, within limitations, that HSNIG could have a role in reducing the symptoms and duration of illness in COVID-19 patients. Furthermore, Slapak et al. found that seawater nasal irrigation three times daily for 8 weeks can reduce the occurrence of upper respiratory tract infections compared to no treatment [26]. The authors also reported that seawater nasal irrigation led to a consistent improvement in rhinologic symptoms and milder disease and was associated with decreased medication consumption and a lower rate of complications [26].

In a previous pilot trial published by our group, we have demonstrated that nasal washes with normal saline effectively decrease SARS-CoV-2 viral load within 24 h after admission and at 14 days post-discharge in patients with COVID-19 [6]. Interestingly, Yildiz et al. have demonstrated that the addition of a spray containing the glucocorticoid triamcinolone acetonide to the regimen of saline irrigation was more effective than no treatment and saline irrigation alone in treating olfactory dysfunction in COVID-19 patients [27].

Similar findings to our current work have been reported in one non-randomized study using the same hypertonic seawater solution. Of note, that study also reported symptom relief in the group that performed hypertonic nasal irrigations [28]. Additionally, another study was conducted using a different sterile hypertonic solution which contained seawater, panthenol, xylitol and lactic acid [29]. The authors reported that irrigation shortened the duration of viral shedding by two days without any adverse events [29]. The results of the aforementioned studies and the present trial demonstrate that nasal irrigation with hypertonic seawater solution containing algal and herbal natural ingredients may be more effective than normal saline only. Our results are further supported by a recent study investigating the effect of irrigations with 3% hypertonic saline in adults infected with omicron variant of COVID-19 after adjustment for comorbidities, smoking history, lymphocyte count and Ct values of N gene [30]. These authors also reported that nasal irrigations were associated with quicker reductions in their nucleic acid levels compared to the controls, but there was no noticeable change in the duration of their symptoms [30].

Apart from isotonic and hypertonic saline solutions, researchers have also suggested and evaluated the use of other solutions in COVID-19 patients [31]. In 2021, Yilmaz et al. used a hypertonic alkaline nasal irrigation solution and demonstrated a significant decrease in the nasopharyngeal viral load in COVID-19 patients [32]. Baxter et al. compared povidone-iodine and sodium bicarbonate (both diluted in isotonic saline) in high-risk COVID-19 patients not hospitalized. When both groups were compared to a CDC Surveillance Dataset including laboratory-confirmed COVID-19 cases, an 8-fold hospitalization likelihood was found [33]. In a similar manner, another study found that oral rinse and nasal irrigation with 5% sodium bicarbonate in patients with COVID-19 can enhance virus clearance [34]. Povidone-iodine solution was also used by Kamal Arefin with similar results [35]. Interestingly, an article by Cao et al. suggested the use of a new stable form of the highly potent disinfectant chlorine dioxide [31]. In their work, they reviewed the available animal and human research on the topic and concluded that irrigation solutions with concentrations between 25–50 ppm would be appropriate for COVID-19 patients both from a safety as well as from an efficacy standpoint. The authors reported preliminary tolerance results on 5 healthy volunteers and mentioned that a larger scale study is underway.

Our findings, in combination with previous publications, could inform the implementation of nasal irrigations for patients with highly transmissible upper respiratory infections in both the inpatient and outpatient settings. In the process of managing a pandemic, such as the COVID-19 pandemic, or a local epidemic with a high incidence of cases, it is important to have strategies in place to reduce the transmission in the community. Thus, having an established method to decrease the viral load in patients that are managed in an outpatient setting is crucial. Baxter et al. demonstrated the feasibility of teaching and guiding patients to use nasal irrigation devices in the outpatient setting, in their study with 79 COVID-19 patients [33]. Furthermore, the authors mention that such interventions are widely used in settings where quick access to medical care is impractical. The rise of telemedicine could assist in getting in touch with patients, explaining the exact process to be followed and allow for questions. On the other hand, establishing an algorithm for nasal irrigation in the inpatient setting is easier than in the outpatient setting. Healthcare staff can demonstrate the use of the irrigation system to the patients, assist in its use and monitor for possible side effects while the patient is hospitalized.

In both cases, there remain some unanswered questions that could be the topic of interest in future studies. Such questions include but are not limited to, what is the ideal solution to be used (e.g., normal saline, hypertonic saline, nasal saline with povidone iodine or sodium bicarbonate)? What is the ideal frequency of irrigations? Who are the ideal candidates? Which is the ideal device to be used? Is there a difference in efficacy between different upper respiratory infections? In the inpatient setting, personnel discarding the fluid used for irrigation should follow the same precautions as with any other intervention in infected patients. In the outpatient setting, each patient should use their own device, and it is advisable that the sink in which the procedure is performed is cleaned thoroughly afterwards. Finally, it is important that healthcare personnel clearly explain the benefits and safety of the intervention to overcome the understandable reluctance of patients to adopt a new intervention.

The study has some limitations that should be acknowledged and should inform the interpretation of our results. First, the trial was conducted at a single centre, and therefore, the results need to be confirmed in a larger, multi-centre study to increase its generalizability. Second, both host factors (previous infection history) and viral factors (different SARS-CoV-2 variants) may have had a significant impact on the dynamics of viral load, which in turn may have influenced the effectiveness of nasal irrigation [5]. In addition, considering the significant variations in Ct values that are commonly observed across COVID-19 patient groups, it would be important to adjust our results by the corresponding content of the house keeping gene [36]. Another important consideration with respect to the measurement of SARS-CoV-2 viral RNA is the discrimination between infectious and inactivated viral particles so as to decrease false-positive result rates. However, it should be noted that very few studies have reported false positive SARS-CoV-2 polymerase chain reaction results, and in all cases, this was due to contamination by inactivated virus vaccine [37,38]. In addition, the Vircell kit which was used in our study is in compliance with international guidelines, has been evaluated for its specificity and sensitivity, has both endogenous and exogenous controls that are important for the validity of the method and has shown concordant results at low viral loads [39]. Another limitation is that the swabs were not processed in the same RT-PCR run, which could have introduced variability between the test results. Finally, the inclusion of an isotonic saline group would have been ideal to draw more robust conclusions. We suggest that future studies should compare isotonic and hypertonic saline solutions in patients with COVID-19.

## 5. Conclusions

Nasal irrigations with hypertonic seawater solution containing algal and herbal natural ingredients decrease the nasopharyngeal viral load and increase the number of patients testing negative for COVID-19 at 14 days post-discharge. This treatment is safe and can serve as an additional hygiene measure providing extra protection against the transmission of SARS-CoV-2. Further, larger-scale, multi-centre studies should strive to answer questions pertaining to ideal irrigation intervals, duration and effectiveness in different viruses and variants.

## Figures and Tables

**Figure 1 jpm-13-01093-f001:**
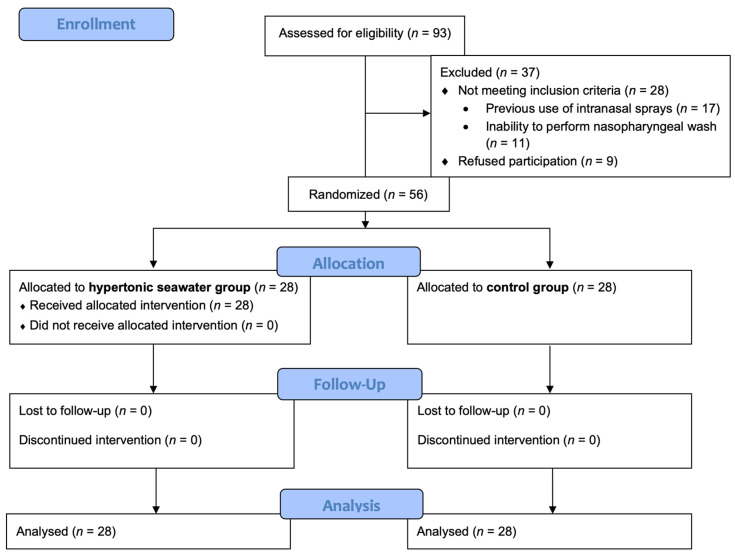
Consort flow diagram of the study.

**Figure 2 jpm-13-01093-f002:**
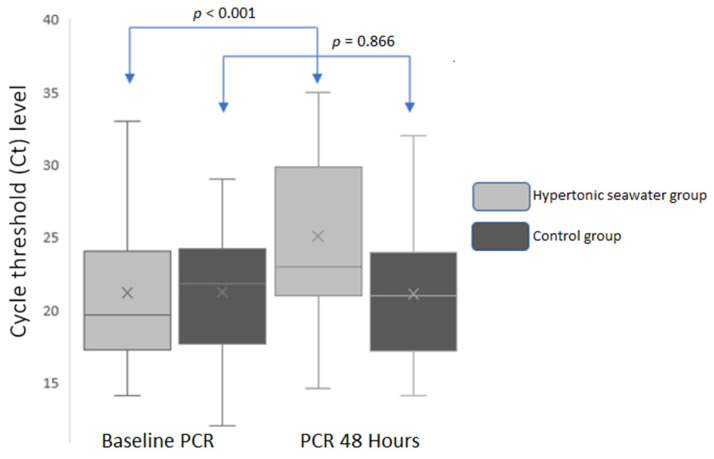
Mean N gene Ct values in baseline and 48 h in both groups.

**Table 1 jpm-13-01093-t001:** Baseline characteristics of the study population.

Characteristics	Hypertonic Seawater Group	Control Group	*p*-Value
Participants, n	28	28	-
Age, years (mean ± SD)	64.1 ± 13	63 ± 14.7	0.53
Male, *n* (%)	20 (71.4)	13 (46.4)	-
Vaccination status	15	16	0.79
PaO_2_/FiO_2_, median (IQR)	333 (281.3–376.5)	358.5 (275.3–388.8)	0.91
Duration of symptoms prior to enrolment, days [median (IQR)]	10.5 (6–16.8)	8.5 (5–14.5)	0.37
Comorbidities			
Coronary disease, *n*	6	7	0.75
Hypertension, *n*	14	12	0.59
Respiratory disease, *n*	5	4	0.72
Obesity, *n*	6	7	0.75
Charlson comorbidity index, median (IQR)	4 (2–5)	4 (1–5)	0.39
Change of taste or smell at enrolment, *n* (%)	8 (28.6)	3 (10.7)	0.09

**Table 2 jpm-13-01093-t002:** Change in SARS-CoV-2 viral load according to vaccination status.

**Study Day**	**Ct Value Baseline**	**Ct Value 48 h**	**ΔCt**	** *p* ** **-Value**
**Hypertonic Seawater Group**				
Vaccinated	20.22 ± 5	24.20 ± 5.78	3.98 ± 2.34	<0.01
Unvaccinated	22.31 ± 4.95	26.04 ± 5.79	3.73 ± 3.77	0.04
**Control Group**				
Vaccinated	20.74 ± 4.88	22.03 ± 5.58	1.28 ± 4.68	0.29
Unvaccinated	21.89 ± 3.99	19.85 ± 3.62	−2.04 ± 2.90	0.03

## Data Availability

The data that support the findings of this study are available from the corresponding author (I.P.), upon reasonable request.

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
