# Peer review of "A Hypertonic Seawater Nasal Irrigation Solution Containing Algal and Herbal Natural Ingredients Reduces Viral Load and SARS-CoV-2 Detection Time in the Nasal Cavity"

_jpm, 2023, doi:10.3390/jpm13071093_

Round 1

Reviewer 1 Report

The manuscript by Pantazopoulos et al described a finding that a hypertonic seawater nasal irrigation solution is helpful to reduce viral load in nasal cavity. Although the finding is interesting, I have some major concerns as follows:

1. The major conclusion is drawn from ΔCt, however, the sampling error is unavoidable. Thus, I recommend to adjust the results by the coressponding content of a house keeping gene.

2. Could the author show the reason why 37 patients were excluded. Is there any subjective trade-off?

Minor revision of the English language is required to make the manuscript more smooth.

Author Response

Detailed response to reviewer’s comments

Reviewer: 1

The manuscript by Pantazopoulos et al described a finding that a hypertonic seawater nasal irrigation solution is helpful to reduce viral load in nasal cavity. Although the finding is interesting, I have some major concerns as follows:

Thank you very much for finding our article interesting and the time you devoted in reviewing this manuscript. We hope that the revised manuscript will satisfy your high standards and you will finally consent to its publication.

  1. The major conclusion is drawn from ΔCt, however, the sampling error is unavoidable. Thus, I recommend to adjust the results by the coressponding content of a house keeping gene.

We very much appreciate this comment. We agree with the reviewer that accurate determination of gene expression requires data normalization with a housekeeping gene thus controlling for experimental variations such as RNA extraction and reverse transcription efficiencies. In our study, viral RNA genome detection was determined using the commercial Vircell Direct SARS-CoV-2 RealTime PCR Kit. Although the latter includes the RNAse P gene as an internal endogenous human control, this has only been validated for the detection of improper sample collection or degradation. We have added this limitation as well as the relevant literature in the Discussion section of the manuscript.

  1. Could the author show the reason why 37 patients were excluded. Is there any subjective trade-off?

Thank you for the comment. In the Consort flow diagram of the study, it is clear that 28 patients did not meet inclusion criteria (17 due to previous use of intranasal sprays and 11 due to inability to perform nasopharyngeal wash) and 9 patients refused participation.

Comments on the Quality of English Language

Minor revision of the English language is required to make the manuscript more smooth.

The paper has been carefully revised by a native English speaker to improve readability.

Reviewer 2 Report

Pantazopoulos and colleagues describe the effect of nasal rinses on the presence of SARS-CoV-2 in the nasal cavity of hospitalized COVID-19 patients. While the rinses did not seem to affect symptoms, they reduced viral loads significantly. The authors conclude, therefore, that nasal irrigations could 'serve as an additional hygiene measure providing extra protection against the transmission of SARS-CoV-2'.

The methodology is similar to the one described in reference 23 by Gangadi et al. However, an effect on symptoms was reported by Gangadi et al. Do the authors have an explanation for this difference? Also highlighted in ref 23 was the rapid effect on smell or taste. Was there any indication that the patients in this study also experienced more rapid improvement of those senses?

The authors should make clear whether the SARS-CoV-2 detection method distinguishes between replicating and inactive virus. Could rinsing result in selective removal of inactive SARS-CoV-2, so that the impact on infectious virus would be limited?

My main concern is the lack of treatment of the control group. It would have been very interesting to see if the differences between treated and controls would have been significantly different if the control group had received an isotonic rinse. I think this should be highlighted as a limitation. Related to the lack of such a control treatment, the discussion on the possible mechanism of the nasal washes (lines 233-250) seems redundant.

Another point of discussion is the applicability of the findings. How likely is it that people with COVID-19 will have regular nasal rinses? So, while the results are scientifically sound, have the authors any suggestion how this would be used in the regular population or perhaps in a healthcare setting? What about the rinse fluid, are special measures needed to discard it? 

Author Response

Detailed response to reviewer’s comments

Reviewer 2

Pantazopoulos and colleagues describe the effect of nasal rinses on the presence of SARS-CoV-2 in the nasal cavity of hospitalized COVID-19 patients. While the rinses did not seem to affect symptoms, they reduced viral loads significantly. The authors conclude, therefore, that nasal irrigations could 'serve as an additional hygiene measure providing extra protection against the transmission of SARS-CoV-2'.

We would like to thank you for the time you devoted in reviewing this manuscript. We hope that the revised manuscript will satisfy you and you will finally consent to its publication.

The primary aim of our study was to investigate the effect of nasal irrigations with a hypertonic seawater solution on nasopharyngeal viral load in hospitalized patients with COVID-19 pneumonia. The secondary objective was to examine if this effect influences escalation to high flow nasal oxygen or non-invasive ventilation, time of detection of SARS-CoV-2 in the nasal mucosa, admission to intensive care unit (ICU), and mortality. We did not examine if the decrease in viral load influences symptoms like dyspnea, cough, fever, sore throat etc. since all patients already had severe COVID-19 pneumonia.

The methodology is similar to the one described in reference 23 by Gangadi et al. However, an effect on symptoms was reported by Gangadi et al. Do the authors have an explanation for this difference?

As I mentioned before the effect on symptoms was not within the objectives of our study. However, the effect on symptoms is debatable since recent studies have shown no significant improvement in symptom disappearance time (Liu L, et al. Effect of Nasal Irrigation in Adults Infected with Omicron Variant of COVID-19: A Quasi-Experimental Study. Front Public Health 2022, 10, 1046112, doi:10.3389/fpubh.2022.1046112). 

Also highlighted in ref 23 was the rapid effect on smell or taste. Was there any indication that the patients in this study also experienced more rapid improvement of those senses?

Thank you for the comment. We did not examine if the decrease in viral load influenced the smell or taste. 

The authors should make clear whether the SARS-CoV-2 detection method distinguishes between replicating and inactive virus. Could rinsing result in selective removal of inactive SARS-CoV-2, so that the impact on infectious virus would be limited?

We very much appreciate this comment. We agree with the reviewer that it is important to discriminate between replicating and inactivated viral particles so as to decrease false-positive result rates. However, it should be noted that very few studies have reported false positive SARS-CoV-2 RT-PCR results and in all cases, this was due to contamination by inactivated virus vaccine. In addition, the Vircell kit which was used in our study, has been evaluated for its specificity and sensitivity, in compliance with international guidelines and has both endogenous and exogenous controls that are important for the validity of the tests. We have added this limitation as well as the relevant literature in the Discussion section of the manuscript.

My main concern is the lack of treatment of the control group. It would have been very interesting to see if the differences between treated and controls would have been significantly different if the control group had received an isotonic rinse.

Thank you very much for the comment. We totally agree. Our team has previously demonstrated that nasal washes with normal saline effectively decrease SARS-CoV-2 viral load within 24 hours after admission and at 14 days post-discharge in patients with COVID-19. Now that we have demonstrated that a hypertonic seawater nasal irrigation solution is effective in reducing viral load and SARS-CoV-2 detection time in the nasal cavity, the next step is to perform a study comparing this solution with an isotonic saline. However, we will need to recruit many more patients in order to achieve a power of 0.9.

I think this should be highlighted as a limitation. Related to the lack of such a control treatment, the discussion on the possible mechanism of the nasal washes (lines 233-250) seems redundant.

Thank you. We have included a relevant comment in the limitations section.

Another point of discussion is the applicability of the findings. How likely is it that people with COVID-19 will have regular nasal rinses? So, while the results are scientifically sound, have the authors any suggestion how this would be used in the regular population or perhaps in a healthcare setting? What about the rinse fluid, are special measures needed to discard it?

Thank you very much for your insightful comment. Two new paragraphs have been added in the discussion section addressing the topics that you suggested.